



# Improvement of the Rnnmm type climate index approach with a spatio-temporal model based on the Hawkes process

Fidel Ernesto Castro Morales[1], Antonio Marcos Batista do Nascimento[1], Marina Silva Paez[2], Daniele Torres Rodrigues[3], and Carla de Moraes Apolinário[1]

[1]Department of Statistics, Federal University of Rio Grande do Norte, Campus Universitário - Lagoa Nova, Natal, 59078-970, Rio Grande do Norte, Brazil.
[2]Department of Statistical Methods, Institute of Mathematics, Federal University of Rio de Janeiro, Avenida Athos da Silveira Ramos, 149 - Edifício do Centro de Tecnologia, Bloco C (Térreo) - Cidade Universitária, Rio de Janeiro, 21941-909, Rio de Janeiro, Brazil.
[3]Department of Statistics, Federal University of Piauí, Av. Universitária, s/n - Ininga, Teresina, 64049-550, Piauí, Brazil.

**Correspondence:** Fidel Ernesto Castro Morales (fidel.castro@ufrn.br)

**Abstract.** This paper proposes an innovative geostatistical model based on self-exciting Hawkes processes for modeling the Rnnmm-type extreme climate index, representing a novel contribution to the literature on climate extremes. The proposed approach generalizes non-homogeneous spatio-temporal Poisson models by incorporating temporal dependence between events through excitation functions, enabling the capture of clustering patterns commonly observed in precipitation time series. The

5 model is formulated within a Bayesian framework, with parameter estimation performed via Markov Chain Monte Carlo (MCMC) methods. Spatial dependence is introduced through hierarchical Gaussian processes, allowing for interpolation in locations without observed data. The model is applied to the R20mm index (annual number of days with precipitation exceeding 20 mm) using data from northern Maranhão (Brazil) for the period 2013–2022. Cross-validation results demonstrate that the proposed model outperforms non-homogeneous Poisson models with and without seasonality in terms of predictive accuracy.

Furthermore, the excitation parameters provide additional insights into the persistence and intensity of extreme events, revealing patterns not captured by conventional models. These findings highlight the model's potential to enhance the analysis of climate extremes in regions with high spatio-temporal variability in precipitation.

## 1 Introduction

The occurrence of extreme climate events, such as intense and prolonged rainfall, poses significant challenges in vulnerable regions, particularly in areas characterized by high climate variability. In the state of Maranhão, Brazil, for instance, the rainfall regime exhibits pronounced seasonality. During the rainy season, precipitation events exceeding 20 mm tend to occur frequently, with short intervals between them. As the dry season approaches, these intervals gradually increase until such events



cease to be recorded. Capturing this dynamic and irregular behavior is essential for a more accurate understanding of climate
extremes and for supporting mitigation and adaptation strategies in local contexts.

In this regard, climate indices developed by the Expert Team on Climate Change Detection and Indices (ETCCDI), such as the Rnnmm index, have proven to be valuable tools for measuring and monitoring climate extremes. The Rnnmm index is defined as the annual number of days in which daily precipitation exceeds nn mm, where nn is a user-defined threshold. Despite its usefulness, the statistical modeling of these indices still faces limitations, especially in regions affected by large-
scale phenomena such as El Niño and La Niña, which irregularly influence seasonal precipitation cycles.

Recent studies, such as Morales and Vicini (2020) and Morales and Rodrigues (2023), have investigated the behavior of extreme rainfall frequencies using the Rnnmm index. Morales and Vicini (2020) incorporated anisotropy into the spatial co-variance structure of a spatiotemporal model based on inhomogeneous Poisson processes, originally proposed by Morales et al. (2017), thereby enhancing the model's ability to represent complex spatial patterns. Subsequently, Morales and Rodrigues
(2023) developed a more comprehensive model that accounts for the high spatial and temporal variability of precipitation and captures regular seasonal rainfall cycles by incorporating a cyclic function into the intensity function. However, despite these advances, the approach proposed by Morales and Rodrigues (2023) has limitations in capturing temporal factors that influence the index, particularly in regions such as northeastern Brazil, where precipitation cycles are strongly affected by irregular large-scale climate phenomena, such as El Niño and La Niña, leading to significant deviations from expected seasonal patterns.
To address these limitations, Morales (2023) proposed an alternative approach based on partitioning the analysis period, allowing different intensity functions to be defined for each partition of the inhomogeneous Poisson process. This method employs a priori specification of the intensity function parameters using state-space models (West and Harrison, 1997), providing greater flexibility in the modeling process. Additionally, the proposed approach incorporates anisotropy in the spatial covariance structure, further improving the representation of complex spatial patterns. By explicitly considering temporal variability,
this strategy enhances the model's ability to capture fluctuations driven by global climate phenomena, resulting in a more refined understanding of extreme precipitation events over time.

Moreover, one of the major challenges in developing studies in this field is the limited spatial and temporal coverage of historical data, particularly in developing countries such as Brazil, where the density of rain gauges is below the threshold recommended by the World Meteorological Organization (Curtarelli et al., 2014). To overcome this limitation, researchers
have increasingly relied on remote sensing estimates, such as those provided by satellite products. However, studies indicate that these estimates often underestimate extreme precipitation events. For example, Rodrigues et al. (2020) found that the 3B42V7 product from the Tropical Rainfall Measuring Mission satellite underestimated extreme events between 2000 and 2015 in northeastern Brazil. Similarly, dos Santos et al. (2022) analyzed the IMERG product from the Global Precipitation Measurement (GPM) mission and demonstrated that the quality of climate index estimates, including Rnnmm, depends on
factors such as location and time period. Comparable findings were reported by Batista et al. (2024), who evaluated eight extreme precipitation indices estimated by IMERG in the Parnaíba basin.

Although the models proposed by Morales and Vicini (2020); Morales and Rodrigues (2023) and Morales (2023) have advanced the modeling of the Rnnmm index, several aspects remain unexplored. For instance, the specific seasonal patterns of



the rainfall regime in Maranhão have not yet been thoroughly examined. To bridge this gap, this study proposes an innovative geostatistical model based on self-exciting Hawkes processes, which enables a more accurate representation of the temporal and spatial dynamics of climate extremes, particularly in regional contexts such as Maranhão. This approach is crucial for improving the understanding of the frequency and intensity of extreme events, thereby contributing to the development of effective mitigation and adaptation strategies for climate change.

## 2  The Model

We now formally define the geostatistical model considered in this article. Let $\mathcal{A} \subset \mathbb{R}^d$ denote the spatial domain of interest, with $n$ fixed observation sites located at $\boldsymbol{s}_1, \ldots, \boldsymbol{s}_n \in \mathcal{A}$. These sites represent the locations where data are collected over time. The data are assumed to originate from an underlying stochastic process, evolving as follows.

At each site $\boldsymbol{s}_j$, for $j = 1, \ldots, n$, we observe a continuous-time counting process $N_j(t)$, which keeps track of the number of events that have occurred at location $\boldsymbol{s}_j$ up to time $t$. The history of each process is denoted by $\mathcal{H}_{\boldsymbol{\Phi}_j} = \{\mathcal{H}_{\boldsymbol{\Phi}_j}(t), t \geq 0\}$, and it depends on a set of unknown parameters $\boldsymbol{\Phi}_j = (\boldsymbol{\varphi}_j, \boldsymbol{\varrho}_j)$, where $\boldsymbol{\varphi}_j$ and $\boldsymbol{\varrho}_j$ are vectors whose specific roles will be defined later. The collection of all such parameters across sites is denoted by $\boldsymbol{\Phi} = (\boldsymbol{\Phi}_j)_{j=1}^n$.

Conditional on a realization of $\boldsymbol{\Phi}$, we assume that the counting processes $\{N_j(t)\}_{j=1}^n$ evolve independently according to self-exciting point processes, meaning that the occurrence of an event at any given time increases the likelihood of subsequent events occurring in the near future. This characteristic makes the model particularly suitable for representing extreme weather events, where an intense occurrence at a given location tends to trigger additional events within a short period. Specifically, each $N_j(t)$ follows a Hawkes process, a model designed to capture clustering patterns in the event occurrences.

The likelihood of an event occurring at site $\boldsymbol{s}_j$ at time $t$ is governed by the conditional intensity function $\lambda_j(t \mid \boldsymbol{\Phi})$, which is defined as:

$$\lambda_j(t \mid \boldsymbol{\Phi}) = g_j(t \mid \boldsymbol{\Phi}) + \sum_{t_{j,k} < t} \mu_j(t - t_{j,k} \mid \boldsymbol{\Phi}). \tag{1}$$

Here, the function $g_j(t \mid \boldsymbol{\Phi})$ represents a baseline rate of events, independent of past occurrences. This can be interpreted as the background intensity, describing how frequently events would occur if there were no interactions between them. The second term, $\sum_{t_{j,k} < t} \mu_j(t - t_{j,k} \mid \boldsymbol{\Phi})$, accounts for the influence of past events.

The quantity $t_{j,k}$ denotes the occurrence time of the $k$-th event observed at site $\boldsymbol{s}_j$, with all such events forming the ordered sequence $\{t_{j,1}, t_{j,2}, \ldots\}$. Each past event at time $t_{j,k}$ contributes an additional increase to the intensity at time $t$, according to the function $\mu_j(t - t_{j,k} \mid \boldsymbol{\Phi})$. This function, often called the excitation kernel, controls how strongly and for how long past events influence the likelihood of future occurrences.

To precisely characterize this excitation mechanism, we assume that the function $\mu_j(\tau \mid \boldsymbol{\Phi})$ follows an exponential decay:

$$\mu_j(\tau \mid \boldsymbol{\Phi}) = \mu(\tau \mid \boldsymbol{\varrho}_j) = \alpha_j e^{-\beta_j \tau}, \quad \tau \geq 0, \tag{2}$$

where $\boldsymbol{\varrho}_j = (\alpha_j, \beta_j)$, with $\alpha_j > 0$, $\beta_j > 0$, and $\alpha_j < \beta_j$. This formulation ensures that the influence of past events diminishes over time, capturing the natural decay in their impact on future occurrences.



Simultaneously, the background intensity function $g_j(t \mid \mathbf{\Phi})$ is assumed to follow the parametric form:

$$g_j(\tau \mid \mathbf{\Phi}) = g(\tau \mid \boldsymbol{\varphi}_j) = \gamma_j \eta_j \tau^{\eta_j - 1}, \quad \tau \geq 0, \tag{3}$$

where $\boldsymbol{\varphi}_j = (\gamma_j, \eta_j)$ are site-specific parameters that control the baseline occurrence of events independently of past observations. This parametric form provides flexibility in capturing different temporal patterns of extreme events.

The model specified by the excitation function in (2) and the background rate in (3) will be referred to as the *Weibull-Hawkes model*, reflecting the Weibull-distributed baseline hazard function and the Hawkes process governing self-exciting dynamics.

## 2.1 Model's Prior Distributions

To fully specify the Weibull-Hawkes model in a Bayesian framework, we must assign prior distributions to its parameters. These priors reflect our initial beliefs about the parameter values before incorporating observed data and play a fundamental role in the inference process. A well-chosen prior structure not only provides regularization, but also ensures that the model remains flexible enough to capture the underlying patterns in the data.

In this work, we assign prior distributions to the parameters $\beta_j$, $\alpha_j$, $\gamma_j$, and $\eta_j$ for $j = 1, \ldots, n$. The choice of these priors is guided by both computational considerations and prior knowledge about their plausible values.

The excitation parameter $\beta_j$ is particularly relevant, as it controls the decay rate of the self-excitation function and influences the temporal clustering of events. The selection of its prior is crucial for the stability and convergence of the estimation algorithm. Some studies suggest using a uniform prior $U(0, a)$ for a suitable choice of constant $a > 0$ (citation needed). However, in this work, we opt for a hierarchical formulation that introduces additional flexibility while preserving interpretability. Specifically, we assume that $\beta_j$ follows the hierarchical structure:

$$\beta_j = a Z_j, \quad \text{where} \tag{4}$$

$$
\begin{aligned}
Z_j &\sim \text{Beta}(\nu\tau, (1-\tau)\nu), \\
\tau &\sim \text{Beta}(a_\tau, b_\tau),
\end{aligned}
\tag{5}
$$

where $\text{Beta}(\cdot, \cdot)$ denotes the Beta distribution, and $a$, $\nu$, $a_\tau$, and $b_\tau$ are known constants. This hierarchical formulation allows for variability across locations while maintaining control over the range of $\beta_j$ values.

For the background rate $\gamma_j$ and the excitation parameter $\alpha_j$, we assign priors to their logarithmic transformations to ensure positivity and improve numerical stability. Specifically, we define:

$$W_j = \log(\gamma_j), \quad U_j = \log(\alpha_j), \quad M_j = \log(\eta_j), \tag{6}$$





for each $j = 1, \ldots, n$. Instead of assuming independent priors for these transformed parameters, we introduce a spatial dependency structure by modeling them as realizations of Gaussian processes:

$$W(\cdot) \sim \mathcal{GP}(\mathbf{x}_W(\cdot)'\boldsymbol{\Psi}_W, \sigma_W^2 \rho_{\phi_W}(\cdot, \cdot)),$$
$$U(\cdot) \sim \mathcal{GP}(\mathbf{x}_U(\cdot)'\boldsymbol{\Psi}_U, \sigma_U^2 \rho_{\phi_U}(\cdot, \cdot)), \tag{7}$$
$$M(\cdot) \sim \mathcal{GP}(\mathbf{x}_M(\cdot)'\boldsymbol{\Psi}_M, \sigma_M^2 \rho_{\phi_M}(\cdot, \cdot)),$$

where $\mathbf{x}_k$ represents a vector of covariates, $\boldsymbol{\Psi}_k$ is a vector of unknown regression coefficients, $\sigma_k^2$ is a scale parameter, and $\rho_{\phi_k}(\cdot, \cdot)$ is a spatial correlation function depending on the parameter $\phi_k$, for $k \in \{W, U, M\}$. This formulation ensures that sites closer to each other exhibit more similar parameter values, allowing spatial smoothing.

By standard properties of Gaussian processes, the prior distributions of the transformed parameters are given by:

$$\mathbf{W} \sim N(\mathbf{X}_W \boldsymbol{\Psi}_W, \Sigma_W),$$
$$\mathbf{U} \sim N(\mathbf{X}_U \boldsymbol{\Psi}_U, \Sigma_U), \tag{8}$$
$$\mathbf{M} \sim N(\mathbf{X}_M \boldsymbol{\Psi}_M, \Sigma_M),$$

where $\mathbf{X}_k$ is a matrix of covariates, and $\Sigma_k$ is the covariance matrix with entries:

$$\Sigma_k[i,j] = \sigma_k^2 \rho_{\phi_k}(\boldsymbol{s}_i, \boldsymbol{s}_j). \tag{9}$$

To model spatial dependencies, we assume an exponential correlation function for all processes $W$, $U$, and $M$, such that:

$$\rho_{\phi_k}(\boldsymbol{s}_i, \boldsymbol{s}_j) = \exp(-\phi_k |\boldsymbol{s}_i - \boldsymbol{s}_j|), \tag{10}$$

where $|\cdot|$ denotes the Euclidean distance between locations $\boldsymbol{s}_i$ and $\boldsymbol{s}_j$. This choice ensures that correlation decays smoothly as
the distance between sites increases.

To complete the prior specification, we assign hyperpriors to the unknown parameters governing the Gaussian process structure. Specifically, for each $k \in \{W, U, M\}$, we assume:

$$\boldsymbol{\Psi}_k \sim N(\mathbf{m}_k, \mathbf{C}_k),$$
$$\sigma_k^2 \sim \text{Gamma}(a_{\sigma_k}, b_{\sigma_k}), \tag{11}$$
$$\phi_k \sim \text{Gamma}(a_{\phi_k}, b_{\phi_k}),$$

where $\mathbf{m}_k$, $\mathbf{C}_k$, $a_{\sigma_k}$, $b_{\sigma_k}$, $a_{\phi_k}$, and $b_{\phi_k}$ are known hyperparameters that control the mean, variance, and spatial correlation
range of each Gaussian process.

To formally describe the full prior distribution of the Weibull-Hawkes model, let:

$$\boldsymbol{\Theta} = (\boldsymbol{\beta}, \mathbf{W}, \mathbf{U}, \mathbf{M}, \boldsymbol{\Psi}_W, \boldsymbol{\Psi}_U, \boldsymbol{\Psi}_M, \sigma_W^2, \sigma_U^2, \sigma_M^2, \phi_W, \phi_U, \phi_M). \tag{12}$$

The joint prior distribution factorizes as:

$$p(\boldsymbol{\Theta}) = p(\boldsymbol{\beta})p(\mathbf{W})p(\mathbf{U})p(\mathbf{M}) \prod_{k=W,U,M} p(\boldsymbol{\Psi}_k)p(\sigma_k^2)p(\phi_k). \tag{13}$$



This factorization assumes that the priors for the excitation parameter vector $\boldsymbol{\beta}$, the transformed background and excitation parameters $\mathbf{W}$, $\mathbf{U}$, and $\mathbf{M}$, and the hyperparameters governing the Gaussian processes are mutually independent. That is, we assume that knowledge about one group of parameters does not inform or constrain the prior distribution of another.

Although the assumption of prior independence simplifies computations and facilitates inference, it may not always be strictly valid in practical applications. In some cases, incorporating dependencies between priors through hierarchical structures or copula models could improve the model's flexibility. However, for the purposes of this work, we adopt the independence assumption to maintain tractability while still allowing for spatial dependencies to be captured through the Gaussian process priors on $\mathbf{W}$, $\mathbf{U}$, and $\mathbf{M}$.

This completes the prior specification for the Weibull-Hawkes model, establishing a structured Bayesian framework that integrates parameter uncertainty while accounting for spatial dependencies.

## 3  Bayesian Inference

In this section, we outline the Bayesian inference procedure used to estimate the parameters of the Weibull-Hawkes model. Our objective is to derive the posterior distribution of the parameter set $\boldsymbol{\Theta}$, given the observed event times $\boldsymbol{t}$.

Let $L(\boldsymbol{\Theta} \mid \boldsymbol{t})$ denote the likelihood function of $\boldsymbol{\Theta}$, conditional on the observed data $\boldsymbol{t}$. Due to the model's construction, the likelihood factorizes across the $n$ observation sites as:

$$L(\boldsymbol{\Theta} \mid \boldsymbol{t}) = \prod_{j=1}^{n} L_j(\boldsymbol{\Theta} \mid \boldsymbol{t}_j), \tag{14}$$

where $L_j(\boldsymbol{\Theta} \mid \boldsymbol{t}_j)$ represents the likelihood contribution from site $\boldsymbol{s}_j$. The likelihood at each site is given by:

$$L_j(\boldsymbol{\Theta} \mid \boldsymbol{t}_j) = \left[ \prod_{i=1}^{m_j} \lambda_j(t_{j,i} \mid \boldsymbol{\Theta}) \right] \exp(-\Lambda_j(T_j \mid \boldsymbol{\Theta})), \tag{15}$$

where $\lambda_j(t \mid \boldsymbol{\Theta})$ denotes the conditional intensity function of the process at site $\boldsymbol{s}_j$, evaluated at time $t$, and $\Lambda_j(T \mid \boldsymbol{\Theta})$ represents the integrated intensity over the observation window $[0, T]$, corresponding to the expected number of events up to time $T$.

For the Weibull-Hawkes model, the conditional intensity function is given by:

$$\lambda_j(t \mid \boldsymbol{\Theta}) = \gamma_j \eta_j t^{\eta_j - 1} + \sum_{t_{j,i} < t} \alpha_j e^{-\beta_j(t - t_{j,i})}, \tag{16}$$

where the first term corresponds to the background Weibull rate, while the second term captures the influence of past events at site $\boldsymbol{s}_j$. The integrated intensity is expressed as:

$$\Lambda_j(T_j \mid \boldsymbol{\Theta}) = \gamma_j T_j^{\eta_j} + \frac{\alpha_j}{\beta_j} \sum_{k=1}^{n_j(T_j)} \left( 1 - e^{-\beta_j(T_j - t_{j,k})} \right), \tag{17}$$

where $n_j(T_j)$ denotes the number of observed events at site $\boldsymbol{s}_j$ up to time $T_j$.





Bayesian inference is conducted by combining the likelihood function with the prior distribution to obtain the posterior distribution of the parameters. Using Bayes' theorem, the posterior is proportional to:

$$p(\boldsymbol{\Theta} \mid \boldsymbol{t}) \propto L(\boldsymbol{\Theta} \mid \boldsymbol{t})p(\boldsymbol{\Theta}), \tag{18}$$

where $p(\boldsymbol{\Theta})$ is the prior distribution specified in the previous section.

Given the complexity of the likelihood function and the hierarchical structure of the model, exact analytical inference is intractable. Therefore, we employ Markov chain Monte Carlo (MCMC) methods to sample from the posterior distribution. Specifically, a Metropolis-Hastings algorithm or a Gibbs sampler can be used to efficiently explore the high-dimensional posterior space. These methods not only provide point estimates for the parameters but also allow for uncertainty quantification, yielding credible intervals for key parameters such as $\beta_j$, $\alpha_j$, $\gamma_j$, and $\eta_j$ in the Weibull-Hawkes model.

## 170 3.1 Estimation Scheme

We employ a Markov Chain Monte Carlo (MCMC) algorithm to estimate the model parameters. The estimation process iteratively updates the parameters by sampling from their respective full conditional distributions. To ensure reliable posterior inference, we discard the initial samples (burn-in period) and retain only the post-convergence samples for posterior estimation.

  The full estimation scheme is detailed in Algorithm 1.





---

**Algorithm 1** MCMC Sampling Scheme for Parameter Estimation

---

**Input:** Initial values $\boldsymbol{\Theta}^{(0)}$, total number of iterations $K$, burn-in period $B$

1 **for** $k = 1$ *to* $K$ *(total iterations)* **do**

    /* Step 1: Sample the regression coefficients                                                      */

2     **for** $X \in \{W, U, M\}$ **do**

3         Sample $\boldsymbol{\Psi}_X^{(k)}$ from:

4         $\boldsymbol{\Psi}_X^{(k)} \sim N(\mathbf{A}_X, \mathbf{B}_X)$, where $\mathbf{A}_X = (\mathbf{m}_X \mathbf{C}_X^{-1} + \mathbf{X}'\boldsymbol{\Sigma}_X^{-1}\mathbf{X})\mathbf{B}_X$   $\mathbf{B}_X = (\mathbf{C}_X^{-1} + \mathbf{X}'\boldsymbol{\Sigma}_X^{-1}\mathbf{X})^{-1}$

    /* Step 2: Sample the variance parameters                                                     */

5     **for** $X \in \{W, U, M\}$ **do**

6         Sample $\sigma_X^{2\,(k)}$ from:

7         $\sigma_X^{2\,(k)} \sim \mathrm{GI}\left(\frac{n}{2} + a_{\sigma_X}, \frac{1}{2}(\mathbf{X} - \mathbf{X}_X\boldsymbol{\Psi}_X^{(k)})'\mathbf{R}_X^{-1}(\mathbf{X} - \mathbf{X}_X\boldsymbol{\Psi}_X^{(k)}) + b_{\sigma_X}\right)$

    /* Step 3: Sample the spatial dependence parameters                                  */

8     **for** $X \in \{W, U, M\}$ **do**

9         Sample $\phi_X^{(k)}$ from:

10         $P(\phi_X \mid \boldsymbol{\Theta}_{-\phi_X}^{(k)}) \propto \phi_X^{a_{\phi_X}-1}|\boldsymbol{\Sigma}_X|^{-\frac{1}{2}} \times \exp\left[-\frac{1}{2}(\mathbf{X} - \mathbf{X}_X\boldsymbol{\Psi}_X^{(k)})'\boldsymbol{\Sigma}_X^{-1}(\mathbf{X} - \mathbf{X}_X\boldsymbol{\Psi}_X^{(k)}) - b_{\phi_X}\phi_X\right]$

    /* Step 4: Sample the latent variables                                                  */

11     **for** $X \in \{W, U, M\}$ **do**

12         Sample $\mathbf{X}^{(k)}$ from:

13         $p(\mathbf{X} \mid \boldsymbol{\Theta}_{-\mathbf{X}}^{(k)}) \propto L^k(\boldsymbol{\Theta}^k \mid \mathbf{t}) \exp\left[-\frac{1}{2}(\mathbf{X} - \mathbf{X}_X\boldsymbol{\Psi}_X^{(k)})'\boldsymbol{\Sigma}_X^{-1}(\mathbf{X} - \mathbf{X}_X\boldsymbol{\Psi}_X^{(k)})\right]$

    /* Step 5: Sample the auxiliary parameter                                         */

14     **for** $j = 1, \ldots, n$ **do**

15         Sample $\beta_j^{(k)}$ from:

16         $p(\beta_j \mid \boldsymbol{\Theta}_{-\beta_j}^{(k)}) \propto L_j^k(\boldsymbol{\Theta}^k \mid \mathbf{t}_j)\beta_j^{\nu\tau}(a - \beta_j)^{(1-\tau)\nu}$

  /* Step 6: Remove burn-in period                                                              */

17 Discard initial $B$ samples $(\boldsymbol{\Theta}^{(1)}, \ldots, \boldsymbol{\Theta}^{(B)})$

18 Retain samples $\{\boldsymbol{\Theta}^{(B+1)}, \ldots, \boldsymbol{\Theta}^{(K)}\}$ for posterior estimation

**Output:** Posterior samples $\{\boldsymbol{\Theta}^{(B+1)}, \ldots, \boldsymbol{\Theta}^{(K)}\}$ after burn-in

---





## 4 Interpolation

In this section, we address the problem of estimating the conditional intensity function, $\lambda_*(t)$, at a new location $s^*$, where no events have been previously recorded. Since the model assumes that event occurrences are spatially correlated, we can leverage information from the observed locations $\{s_1, s_2, \ldots, s_n\}$ to make predictions at unobserved sites. This is achieved by applying spatial interpolation techniques to the components of the intensity function.

Specifically, we consider interpolation for the background intensity function $g(\tau|\varphi)$ and the excitation function $\mu(\tau \mid \varrho)$. In Section 4.1, we outline the procedure for interpolating $g(\tau|\varphi)$ at the new location; in Section 4.2, we extend the interpolation framework to $\mu(\tau \mid \varrho)$, ensuring that excitation dynamics are spatially consistent; finally, in Section 4.3, we describe how these interpolated components are combined to estimate the full conditional intensity function $\lambda_*(t)$ over continuous time.

### 4.1 Interpolation of the function $g(\tau|\varphi)$

At the new location $s^*$, the background intensity function is given by $g(\tau \mid \varphi_*) = \gamma_* \eta_* \tau^{\eta_* - 1}$, where the parameter vector $\varphi_* = (\gamma_*, \eta_*)$ must be inferred from the observed locations $\{\varphi_1, \varphi_2, \ldots, \varphi_n\}$.

Recall that we defined $W_j = \log(\gamma_j)$ and $M_j = \log(\eta_j)$ as the logarithmic transformations of $\gamma_j$ and $\eta_j$. Since these transformed parameters follow Gaussian process priors, the joint distribution of the observed values $W_1, W_2, \ldots, W_n$ and the unobserved value $W_*$ at $s^*$ is Gaussian. The same holds for $M_*$.

The posterior predictive distribution for $W_*$ is given by:

$$\mathbf{W}_* \mid W_1, \ldots, W_n \quad \sim \quad N(\mu_*, \Sigma_*), \text{ where} \tag{19}$$

$$\mu_* = \mathbf{x}_W(s_*)' \mathbf{\Psi}_W + \Sigma_W(s_*, s_{1:n}) \Sigma_W^{-1} (\mathbf{W} - \mathbf{x}_W(s_{1:n})' \mathbf{\Psi}_W), \tag{20}$$

$$\Sigma_* = \Sigma_W(s_*, s_*) - \Sigma_W(s_*, s_{1:n}) \Sigma_W^{-1} \Sigma_W(s_*, s_{1:n})'. \tag{21}$$

A similar expression holds for $\mathbf{M}_*$, where $\Sigma_M$ and $\mathbf{x}_M$ replace $\Sigma_W$ and $\mathbf{x}_W$, respectively.

At each iteration of the MCMC algorithm, after drawing samples of $\mathbf{W}_*$ and $\mathbf{M}_*$, we obtain the interpolated parameters $\gamma^* = \exp(\mathbf{W}^*)$ and $\eta^* = \exp(\mathbf{M}^*)$ at the new location $s^*$, conditioned on the observed values $\mathbf{W}$ and $\mathbf{M}$. Consequently, in each iteration, we compute the estimated background intensity function $g(t \mid \varphi^*)$ at $s^*$, allowing us to extend the model to locations where no events have been previously recorded.

### 4.2 Interpolation of the function $\mu(\tau \mid \varrho)$

Interpolating the excitation function $\mu(\tau \mid \varrho)$ presents a unique challenge compared to the background intensity function $g(\tau \mid \varphi)$. Unlike $g(\tau \mid \varphi)$, which depends on continuous parameters that can be directly interpolated using Gaussian processes, $\mu(\tau \mid \varrho)$ is event-driven. It depends on the discrete event times at each location, making it more sensitive to local temporal clustering patterns. As a result, standard interpolation techniques are not directly applicable, since neighboring locations may exhibit different past event histories, affecting the excitation dynamics.





To address this issue, we introduce a spatially weighted sampling method that probabilistically selects nearby locations, ensuring that the interpolated excitation function at an unobserved site reflects the rainfall dynamics of its surroundings. The key idea is that locations close to each other tend to share similar excitation structures. Therefore, by borrowing information from neighboring sites while incorporating spatial variability, we can construct a reasonable estimate of $\mu(\tau \mid \boldsymbol{\varrho})$ at unobserved locations.

The method consists of two main steps, which are detailed in Algorithm 2: 1. *Spatially weighted selection of reference locations:* A location is chosen from the set of observed sites, with a probability that decreases with distance from the interpolation site $\mathbf{s}^*$. This ensures that closer locations contribute more to the estimate of $\mu(\tau \mid \boldsymbol{\varrho})$ at $\mathbf{s}^*$, while more distant locations have a lower influence. 2. *Reconstruction of the excitation function:* Once a reference location is selected, its event times are used to reconstruct the excitation function at $\mathbf{s}^*$, incorporating the sampled parameters from the MCMC iterations.

To formalize the procedure, let $\mathbf{S}$ be an $n \times 2$ matrix representing the spatial coordinates of the $n$ monitoring stations, and let $A_n = \{1, \ldots, n\}$ be the set of indices corresponding to the rows of $\mathbf{S}$. Given a new location $\mathbf{s}^*$ where we wish to interpolate $\mu_*(\tau \mid \boldsymbol{\varrho})$, we proceed as follows at each iteration of the MCMC algorithm:

---

**Algorithm 2** Sampling from $\mu_*(\tau \mid \boldsymbol{\varrho}^{(k)})$ at an interpolation location $\mathbf{s}^*$

**Input:** Set of monitoring stations $\mathbf{S}$, distance threshold $R$, weighting parameter $q$

```
/* Step 1: Spatially weighted selection of a reference location          */
```

**19** Define the set of candidate reference locations:

$$A_n^* = \{j \in A_n; \|\mathbf{s}^* - \mathbf{S}[j,]\| \leq R\}.$$

Assign weights to each candidate location based on inverse distance weighting:

$$P_j = \frac{\|\mathbf{s}^* - \mathbf{S}[j,]\|^{-q}}{\sum_{i \in A_n^*} \|\mathbf{s}^* - \mathbf{S}[i,]\|^{-q}}.$$

Sample a reference location $\mathbf{s}_j$ from $A_n^*$ with probability proportional to $P_j$.

```
/* Step 2: Estimation of the excitation function μ*                      */
```

**20** Let $\{t_{j,k}\}_{k=1}^{n_j}$ be the set of observed event times at the sampled reference location $\mathbf{s}_j$. Compute the excitation function at $\mathbf{s}^*$ using:

$$\mu_*(\tau \mid \boldsymbol{\varrho}_j^{(k)}) = \sum_{k=1}^{n_j} \alpha_j^{(k)} e^{-\beta_j^{(k)}(\tau - t_{j,k})},$$

for $\tau > t_{j,k}$.

**Output:** Updated value of $\mu_*(\tau \mid \boldsymbol{\varrho}^{(k)})$ for interpolation at $\mathbf{s}^*$

---





As outlined in Algorithm 2, this approach ensures that the interpolated excitation function preserves both spatial and temporal structure in rainfall events. The weighting scheme favors closer locations, capturing local variability while preventing excessive influence from distant sites. Additionally, the exponential decay function naturally reflects the temporal influence of past events, maintaining consistency in excitation dynamics across space.

### 4.3 Interpolation of the function $\lambda_*(t)$

With the interpolated background intensity function $g_*(\tau \mid \boldsymbol{\varphi}_*)$ and excitation function $\mu_*(\tau \mid \boldsymbol{\varrho}_*)$ at the new location $\boldsymbol{s}^*$, we now estimate the full conditional intensity function $\lambda_*(t)$. Since $\lambda_*(t)$ governs the occurrence rate of events at $\boldsymbol{s}^*$ over time, its interpolation must reflect both the spatial structure of the background intensity and the temporal clustering effects captured by the excitation function.

Following the model specification, the conditional intensity function at $\boldsymbol{s}^*$ is given by:

$$\lambda_*(t \mid \boldsymbol{\varphi}_*, \boldsymbol{\varrho}_*) = g_*(t \mid \boldsymbol{\varphi}_*) + \sum_{k=1}^{n_*} \mu_*(t - t_{*,k} \mid \boldsymbol{\varrho}_*), \tag{22}$$

where:

- $g_*(t \mid \boldsymbol{\varphi}_*) = \gamma_* \eta_* t^{\eta_* - 1}$ is the interpolated background intensity function at $\boldsymbol{s}^*$.

- $\{t_{*,k}\}_{k=1}^{n_*}$ represents the set of inferred past event times at $\boldsymbol{s}^*$.

- $\mu_*(t - t_{*,k} \mid \boldsymbol{\varrho}_*) = \alpha_* e^{-\beta_*(t - t_{*,k})}$ is the interpolated excitation function, incorporating the influence of past events at $\boldsymbol{s}^*$.

Since no direct event observations exist at $\boldsymbol{s}^*$, we must infer the event history $\{t_{*,k}\}_{k=1}^{n_*}$. A natural approach is to simulate event times from the interpolated intensity function itself. At each iteration of the MCMC algorithm, we follow these steps:

1. *Draw samples of the background and excitation parameters*: Using the previously computed interpolations, obtain $\boldsymbol{\varphi}_*$ and $\boldsymbol{\varrho}_*$ for the new location $\boldsymbol{s}^*$.

2. *Simulate event times at $\boldsymbol{s}^*$*: Generate a set of candidate event times $\{t_{*,k}\}_{k=1}^{n_*}$ from a Poisson process with intensity $\lambda_*(t \mid \boldsymbol{\varphi}_*, \boldsymbol{\varrho}_*)$.

3. *Compute $\lambda_*(t)$ at each sampled time*: Using Equation (22), evaluate the interpolated conditional intensity function at each time step.

By iterating over this procedure during the MCMC sampling process, we obtain a probabilistic estimate of $\lambda_*(t)$ at the unobserved location, allowing us to characterize the expected occurrence rate of extreme events in regions without direct observations.

This framework ensures that $\lambda_*(t)$ maintains consistency with the spatiotemporal structure of the observed data. The background intensity function preserves large-scale spatial patterns, while the excitation function captures local clustering behavior, ensuring that interpolated event dynamics remain realistic.



## 5 Modeling Extreme Precipitation Events (R20mm) in the Northern Region of Maranhão State

Understanding the occurrence and intensity of extreme precipitation events is crucial for hydrological planning and disaster
mitigation. In this study, we model the frequency and spatial distribution of extreme daily precipitation events, defined as days
with rainfall exceeding 20 mm (R20mm), in the northern region of Maranhão State, Brazil.

To achieve this, we analyze daily accumulated precipitation data (mm) over a 10-year period, from January 1, 2013, to
December 31, 2022. These data were obtained from 20 rain gauges located at national meteorological stations, managed by
the National Institute of Meteorology (INMET) and the National Water and Basic Sanitation Agency (ANA). The datasets are
publicly available through the INMET meteorological database (https://bdmep.inmet.gov.br) and the ANA open data portal
(https://dadosabertos.ana.gov.br).

Figure 1 presents the spatial distribution of the rain gauges (P) used in this study. The study area spans parts of the states
of Maranhão and Piauí, in northeastern Brazil, a region marked by significant climatic and geographical diversity (Moura and
Shukla, 1981; Vale et al., 2024) and persistent socio-economic challenges (IBGE, 2024). Covering approximately 145,611 km$^2$,
it extends between latitudes 3.2ºS and 6.5ºS and longitudes 41.9ºW and 45.5ºW. The region's topography varies considerably,
with elevations ranging from below 25 meters to over 275 meters, influencing local precipitation patterns. According to the
National Institute of Meteorology, annual rainfall in this area ranges from 1000 to 1800 mm (INMET, 2025), with extreme
daily events occasionally exceeding 100 mm (Rodrigues et al., 2020).

Precipitation in the region is governed by multiple atmospheric systems throughout the year. The primary driver of rainfall
is the Intertropical Convergence Zone (ITCZ), which migrates towards northern and northeastern Brazil during summer and
autumn, generating intense precipitation between February and May (Uvo, 1989; Utida et al., 2019). Additionally, the Upper
Tropospheric Cyclonic Vortex (UTCV) plays a significant role, particularly during the summer months from December to
February, further shaping the regional precipitation regime (Kousky and Alonso Gan, 1981; Lyra et al., 2020).

### 5.1 Application and Predictive Performance Analysis

The proposed model is applied to analyze the spatiotemporal dynamics of extreme precipitation events in the northern region
of Maranhão. By estimating its parameters, we aim to characterize the frequency, intensity, and clustering patterns of these
events, providing insights into their underlying drivers. This analysis allows us to better understand the precipitation regime in
the region and assess how extreme rainfall events are distributed over time and space. To ensure the robustness of our approach,
we compare its performance with alternative models commonly used in the literature, evaluating their ability to capture the
observed precipitation dynamics.

The following models are considered for comparison:

- **Model A**: The proposed Hawkes process model.

- **Model B**: A Poisson model with a seasonal component.

- **Model C**: A standard Poisson model without seasonality.





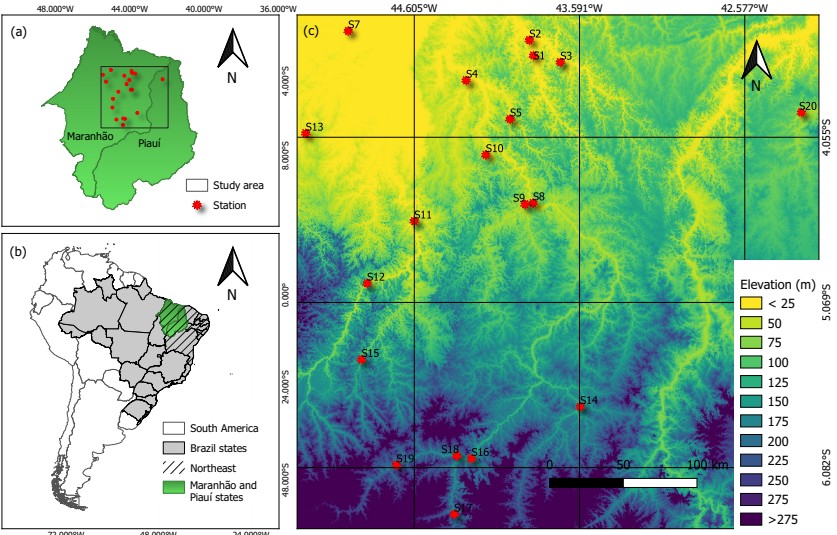

**Figure 1.** Study area and location of rainfall stations.

All three models employ a Weibull intensity function. The processes $W$, $M$, and $U$ incorporate the covariates $X_W = (1, \text{Latitude}, \text{Longitude})$, with $X_M = X_U = X_W$. The variance parameters $\sigma_W^2$, $\sigma_M^2$, and $\sigma_U^2$ follow an Inverse-Gamma prior distribution, IG$(0.001, 0.001)$, while the scale parameters $\phi_W$, $\phi_M$, and $\phi_U$ follow a Gamma prior distribution, G$(0.001, 0.001)$.

In the case of the Hawkes process model, the excitation decay parameter $\beta$ is defined as $\beta = aZ$, where $a = 2$ and $Z \sim \text{Beta}(2\tau, (1 - \tau) \cdot 2)$, allowing for flexibility in capturing self-excitation effects.

Finally, for models incorporating seasonal effects, the prior distributions for the seasonal component parameters $\delta$ and $f$ are given by:

$$p(\delta) = \frac{1}{\sqrt{\delta(100 - \delta)}}, \quad p(f) = \frac{1}{\sqrt{\left(f - \frac{1}{365+10}\right)\left(\frac{1}{365-10} - f\right)}}.$$

This comparative analysis enables us to evaluate how well the proposed model represents precipitation extremes in Maranhão and how it performs relative to other modeling strategies. By contrasting its predictive capabilities with existing methodologies, we assess its effectiveness in capturing the temporal and spatial structure of extreme rainfall events, ensuring a comprehensive understanding of the region's precipitation patterns.

### 5.1.1 Sensitivity Analysis of the Interpolation Method

Before comparing the predictive performance of the models, we conduct a sensitivity analysis to evaluate the impact of key parameters in the interpolation process for the function $\Lambda$. Specifically, we assess the influence of the radius $R$ used to define neighboring stations and the weight vector $P$ assigned to each neighbor.

To analyze the effect of $R$, we consider three different definitions:



- $R_1$: The maximum distance between the observed stations.

- $R_2$: The midpoint between the maximum and average distances.

- $R_3$: The average distance between stations.

Similarly, we test three different weight functions for the neighboring stations:

- $P_{1i} = \frac{|\mathbf{s}_i - \mathbf{s}_k|^{-1}}{\sum_k |\mathbf{s}_i - \mathbf{s}_k|^{-1}}$

- $P_{2i} = \frac{|\mathbf{s}_i - \mathbf{s}_k|^{-3}}{\sum_k |\mathbf{s}_i - \mathbf{s}_k|^{-3}}$

- $P_{3i} = \frac{|\mathbf{s}_i - \mathbf{s}_k|^{-6}}{\sum_k |\mathbf{s}_i - \mathbf{s}_k|^{-6}}$

where $\mathbf{s}_i$ represents the location where the function $\Lambda(\mathbf{s}_i, t)$ is being predicted, and $\mathbf{s}_k$ are the neighboring stations within the
selected radius $R_j$, for $j = 1, 2, 3$.

Tables 1, 2, and 3 present the MAD and MSE values obtained in the cross-validation study for each excluded station ($\mathbf{s}_j$, with $j = 1, \ldots, 20$), considering Model A applied to each radius ($R_1$, $R_2$, and $R_3$) and the different weight vectors ($\mathbf{P}_1$, $\mathbf{P}_2$, $\mathbf{P}_3$). The results indicate that the combination of $R_3$ with $\mathbf{P}_1$ yielded the lowest MAD and MSE values in most cases, leading to its selection for use in the interpolation method of Model A.





**Table 1.** MAD and MSE values obtained in the cross-validation study for each excluded station ($\mathbf{s}_j$, with $j = 1, \ldots, 20$), generated by Model A with radius $R_1$ and different weight vectors ($\mathbf{P}_1$, $\mathbf{P}_2$, $\mathbf{P}_3$).

| | $R_1$ | | | | | |
| | $\mathbf{P}_1$ | | $\mathbf{P}_2$ | | $\mathbf{P}_3$ | |
| Station | MAD | MSE | MAD | MSE | MAD | MSE |
|---|---|---|---|---|---|---|
| $s_1$ | 17 | 316 | 21 | 487 | 21 | 498 |
| $s_2$ | 2 | 7 | 3 | 12 | 3 | 10 |
| $s_3$ | 5 | 51 | 4 | 22 | 4 | 29 |
| $s_4$ | 5 | 34 | 6 | 53 | 6 | 53 |
| $s_5$ | 5 | 31 | 10 | 120 | 14 | 227 |
| $s_6$ | 18 | 493 | 12 | 212 | 16 | 304 |
| $s_7$ | 22 | 624 | 13 | 207 | 9 | 107 |
| $s_8$ | 7 | 60 | 10 | 114 | 10 | 114 |
| $s_9$ | 18 | 346 | 17 | 319 | 17 | 317 |
| $s_{10}$ | 20 | 535 | 16 | 339 | 14 | 251 |
| $s_{11}$ | 6 | 58 | 7 | 64 | 12 | 251 |
| $s_{12}$ | 19 | 461 | 7 | 64 | 11 | 149 |
| $s_{13}$ | 11 | 164 | 21 | 473 | 23 | 599 |
| $s_{14}$ | 28 | 949 | 18 | 374 | 13 | 203 |
| $s_{15}$ | 11 | 154 | 4 | 21 | 4 | 21 |
| $s_{16}$ | 3 | 12 | 2 | 9 | 2 | 9 |
| $s_{17}$ | 6 | 59 | 2 | 8 | 2 | 9 |
| $s_{18}$ | 6 | 41 | 5 | 39 | 5 | 38 |
| $s_{19}$ | 15 | 252 | 9 | 90 | 8 | 79 |
| $s_{20}$ | 43 | 2370 | 49 | 3176 | 54 | 3843 |





**Table 2.** MAD and MSE values obtained in the cross-validation study for each excluded station ($\mathbf{s}_j$, with $j = 1, \ldots, 20$), generated by Model A with radius $R_2$ and different weight vectors ($\mathbf{P}_1$, $\mathbf{P}_2$, $\mathbf{P}_3$).

| | \multicolumn{6}{c}{$R_2$} | | | | | |
| | $\mathbf{P}_1$ | | $\mathbf{P}_2$ | | $\mathbf{P}_3$ | |
| Station | MSA | MSE | MSA | MSE | MSA | MSE |
|---|---|---|---|---|---|---|
| $s_1$ | 18 | 343 | 21 | 487 | 21 | 498 |
| $s_2$ | 3 | 11 | 3 | 12 | 3 | 10 |
| $s_3$ | 4 | 32 | 4 | 21 | 4 | 29 |
| $s_4$ | 4 | 31 | 6 | 54 | 6 | 53 |
| $s_5$ | 5 | 32 | 10 | 119 | 14 | 227 |
| $s_6$ | 13 | 258 | 12 | 193 | 16 | 304 |
| $s_7$ | 18 | 405 | 12 | 191 | 9 | 107 |
| $s_8$ | 7 | 60 | 10 | 114 | 10 | 114 |
| $s_9$ | 18 | 346 | 17 | 319 | 17 | 317 |
| $s_{10}$ | 20 | 535 | 16 | 339 | 14 | 251 |
| $s_{11}$ | 6 | 61 | 7 | 65 | 12 | 252 |
| $s_{12}$ | 19 | 469 | 13 | 223 | 11 | 149 |
| $s_{13}$ | 11 | 181 | 21 | 482 | 23 | 599 |
| $s_{14}$ | 26 | 813 | 17 | 359 | 13 | 204 |
| $s_{15}$ | 12 | 164 | 4 | 21 | 4 | 21 |
| $s_{16}$ | 2 | 9 | 2 | 9 | 2 | 9 |
| $s_{17}$ | 3 | 16 | 2 | 8 | 2 | 9 |
| $s_{18}$ | 5 | 37 | 5 | 39 | 5 | 38 |
| $s_{19}$ | 12 | 171 | 8 | 87 | 8 | 79 |
| $s_{20}$ | 52 | 3531 | 53 | 3695 | 55 | 3938 |





**Table 3.** MAD and MSE values obtained in the cross-validation study for each excluded station ($\mathbf{s}_j$, with $j = 1, \ldots, 20$), generated by Model A with radius $R_3$ and different weight vectors ($\mathbf{P}_1$, $\mathbf{P}_2$, $\mathbf{P}_3$).

| | $R_3$ | | | | | |
|---|---|---|---|---|---|---|
| | $\mathbf{P}_1$ | | $\mathbf{P}_2$ | | $\mathbf{P}_3$ | |
| Station | MSA | MSE | MSA | MSE | MSA | MSE |
| $s_1$ | 19 | 391 | 21 | 487 | 21 | 498 |
| $s_2$ | 4 | 22 | 3 | 12 | 3 | 10 |
| $s_3$ | 4 | 26 | 4 | 21 | 4 | 29 |
| $s_4$ | 5 | 36 | 6 | 56 | 6 | 54 |
| $s_5$ | 6 | 50 | 10 | 123 | 14 | 227 |
| $s_6$ | 11 | 205 | 12 | 197 | 16 | 305 |
| $s_7$ | 16 | 338 | 12 | 185 | 9 | 106 |
| $s_8$ | 6 | 49 | 10 | 114 | 10 | 114 |
| $s_9$ | 19 | 384 | 17 | 318 | 17 | 317 |
| $s_{10}$ | 17 | 394 | 16 | 330 | 14 | 251 |
| $s_{11}$ | 7 | 72 | 7 | 63 | 12 | 252 |
| $s_{12}$ | 16 | 338 | 13 | 218 | 11 | 146 |
| $s_{13}$ | 15 | 282 | 21 | 498 | 23 | 600 |
| $s_{14}$ | 17 | 357 | 15 | 269 | 13 | 193 |
| $s_{15}$ | 5 | 34 | 3 | 19 | 4 | 21 |
| $s_{16}$ | 3 | 12 | 2 | 9 | 2 | 9 |
| $s_{17}$ | 2 | 8 | 2 | 8 | 2 | 9 |
| $s_{18}$ | 5 | 38 | 5 | 39 | 5 | 38 |
| $s_{19}$ | 9 | 103 | 8 | 85 | 8 | 79 |
| $s_{20}$ | 60 | 4748 | 60 | 4748 | 60 | 4748 |

### 5.1.2 Cross-Validation for Predictive Performance

To assess the predictive performance of the models in interpolating the R20mm index, we conduct a leave-one-out cross-validation study. The procedure consists of systematically removing data from one station $\mathbf{s}_i$ at a time and fitting the models using the remaining stations. This process is repeated for all 20 stations in the dataset.

After fitting each model under these 20 different configurations, we apply interpolation methods to estimate the integrated intensity function $\Lambda_j(t)$ at the removed station $\mathbf{s}_j$. The quality of these predictions is then evaluated using the following error





metrics:

$$\text{MAD}_j = \frac{1}{n_j} \sum_{t \in A_j} \left| n_{j,t} - \hat{\Lambda}_j(t) \right|$$

and

$$\text{MSE}_j = \frac{1}{n_j} \sum_{t \in A_j} \left( n_{j,t} - \hat{\Lambda}_j(t) \right)^2,$$

where $n_{j,t}$ is the observed number of extreme precipitation events in the interval $(0,t]$ at location $\mathbf{s}_j$, and $A_j = \{t_{1,j}, \ldots, t_{n_j,j}\}$ is the set of observed event times at that location.

By combining the sensitivity analysis and the cross-validation study, we aim to ensure that the proposed model not only accurately captures the spatiotemporal structure of extreme precipitation events but also demonstrates superior predictive performance compared to alternative approaches.

In Table 4, we observe that Model A generally exhibited the lowest MAD and MSE values compared to the other models, indicating its superior predictive performance. These results suggest that Model A is the most suitable for capturing the observed data patterns, providing more accurate forecasts. Figure 2 displays the predictions of the function $\Lambda_3(t)$ generated by each model, along with their respective 95% credibility intervals. It is evident from this figure that Model A yields the most consistent predictions, with lower uncertainty and a better fit to the observed data, confirming its superior predictive

performance.

Table 4 also shows that Model A outperforms Model B in terms of predictive accuracy. Model A achieved lower MAD and MSE values in 85% of cases compared to Model B. Additionally, Model A also demonstrated superior performance compared to Model C, achieving lower MAD values in 50% of cases, tying in 20%, and showing higher values in only 30% of instances. These results indicate that, overall, Model A provides better predictive performance than Model C.



**Table 4.** MAD and MSE results in the cross-validation study for each excluded station $\mathbf{s}_j$, $j = 1, \ldots, 20$, generated by each model.

| Station | Model A MAD | Model A MSE | Model B MAD | Model B MSE | Model C MAD | Model C MSE |
|---|---|---|---|---|---|---|
| $s_1$ | 19 | 391 | 11 | 172 | 20 | 498 |
| $s_2$ | 4 | 22 | 17 | 476 | 7 | 78 |
| $s_3$ | 4 | 26 | 21 | 717 | 8 | 98 |
| $s_4$ | 5 | 36 | 18 | 560 | 11 | 169 |
| $s_5$ | 6 | 50 | 12 | 281 | 9 | 113 |
| $s_6$ | 11 | 205 | 15 | 392 | 8 | 114 |
| $s_7$ | 16 | 338 | 32 | 1289 | 6 | 60 |
| $s_8$ | 6 | 49 | 3 | 13 | 8 | 86 |
| $s_9$ | 19 | 384 | 12 | 162 | 11 | 165 |
| $s_{10}$ | 17 | 394 | 29 | 1114 | 18 | 421 |
| $s_{11}$ | 7 | 72 | 8 | 102 | 8 | 88 |
| $s_{12}$ | 16 | 338 | 33 | 1422 | 12 | 205 |
| $s_{13}$ | 15 | 282 | 12 | 221 | 15 | 337 |
| $s_{14}$ | 17 | 357 | 50 | 3126 | 10 | 142 |
| $s_{15}$ | 5 | 34 | 29 | 1061 | 5 | 34 |
| $s_{16}$ | 3 | 12 | 31 | 1288 | 3 | 16 |
| $s_{17}$ | 2 | 8 | 36 | 1831 | 5 | 38 |
| $s_{18}$ | 5 | 38 | 41 | 2117 | 5 | 36 |
| $s_{19}$ | 9 | 103 | 45 | 2418 | 14 | 231 |
| $s_{20}$ | 60 | 4748 | 16 | 309 | 31 | 1163 |

335 Table 5 summarizes the estimated values of the parameters $\phi$, $\boldsymbol{\sigma}^2$, $\boldsymbol{\Psi}_W$, $\boldsymbol{\Psi}_M$, $\boldsymbol{\Psi}_U$, $\delta$, $\theta$, and $f$ for Models A, B, and C. The results indicate notable differences in parameter estimates across models, particularly in the spatial dependence parameters $(\phi_W, \phi_M, \phi_U)$ and the variance components $(\sigma_W^2, \sigma_M^2, \sigma_U^2)$. Table 6 presents the estimates of the shape parameter $\eta_j$ of the Weibull intensity function for each station, showing that Model C generally yields higher mean estimates compared to Models A and B, with relatively narrow credibility intervals across all stations. Table 7 presents the estimates of the scale parameter

340 $\gamma_j$ of the Weibull intensity function for each station, showing that Model B generally yields higher mean estimates compared to Models A and C, with relatively narrow credibility intervals across all stations. Finally, Table 8 reports the estimates of the parameters $\alpha_j$ and $\beta_j$ governing the excitation function $\mu$ in the Hawkes process, specifically for Model A. In our application context, the parameter $\alpha_j$ reflects the instantaneous increase in the intensity function following the occurrence of a new event,





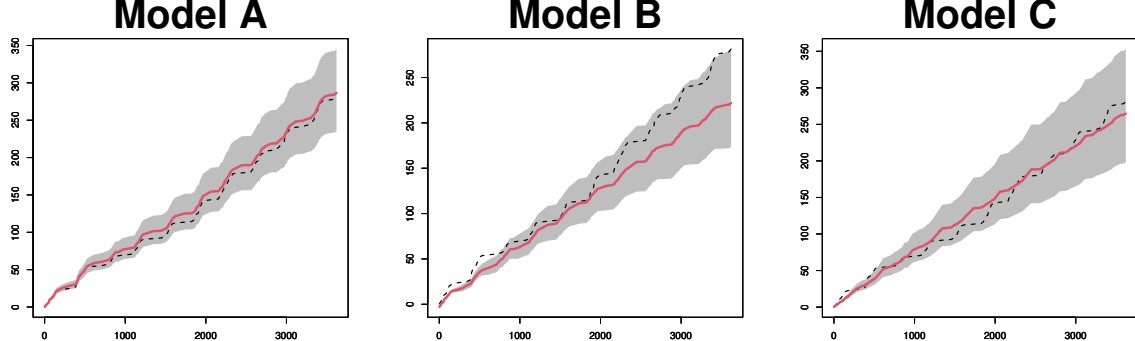

**Figure 2.** Estimated function $\Lambda_3(t)$ (solid red line) with a 95% credibility interval (shaded area) and the observed number of event occurrences in the interval $(0, t]$, $n_{j,t}$ (black dashed line), for each of the analyzed models.

while the parameter $\beta_j$ describes the rate at which the intensity decays back to its baseline level. That is, the occurrence of a
precipitation event exceeding 20 mm in a single day increases the probability of a subsequent similar event by an increment of
0.049 in the intensity, as observed at location $s_1$ (Table 8). This influence decays at a rate of 0.064 (Table 8).

The results suggest some variability in the excitation intensity ($\alpha_j$) and decay rate ($\beta_j$) across different stations, reflecting
the spatial heterogeneity in extreme precipitation events. Higher values of $\alpha_j$ and $\beta_j$ are concentrated in the northern part of
the study area, whereas the lowest values are observed at the southernmost stations ($s_{15}$, $s_{16}$, $s_{17}$, $s_{18}$, and $s_{19}$), where $\alpha_j$
ranges from 0.035 to 0.037, and $\beta_j$ from 0.053 to 0.058 (Table 8). These variations in $\alpha_j$ and $\beta_j$ between the northern and
southern regions may be partly attributed to differences in elevation and rainfall regimes influenced by proximity to the coast
and the activity of the Intertropical Convergence Zone (ITCZ). Northern regions, characterized by lower elevation and higher
atmospheric humidity, tend to exhibit greater intensity and persistence of extreme rainfall events, while southern, more elevated
and inland areas experience lower frequency and weaker clustering of such events.





**Table 5.** Summary of parameter estimates for $\phi$, $\sigma^2$, $\Psi_W$, $\Psi_M$, $\Psi_U$, $\delta$, $\theta$, and $f$ for Models A, B, and C.

| Parameter | Model A | | | Model B | | | Model C | | |
|---|---|---|---|---|---|---|---|---|---|
| | mean | 2.5 | 97.5 | mean | 2.5 | 97.5 | mean | 2.5 | 97.5 |
| $\phi_W$ | 0.076 | 0.005 | 0.496 | 0.009 | 0.005 | 0.023 | 0.043 | 0.005 | 0.299 |
| $\sigma^2_W$ | 0.773 | 0.038 | 4.716 | 0.304 | 0.117 | 0.733 | 0.557 | 0.150 | 1.641 |
| $\phi_M$ | 0.007 | 0.005 | 0.015 | 0.006 | 0.005 | 0.008 | 0.006 | 0.005 | 0.011 |
| $\sigma_M$ | 0.322 | 0.131 | 0.734 | 0.115 | 0.061 | 0.217 | 0.165 | 0.082 | 0.320 |
| $\phi_U$ | 0.029 | 0.005 | 0.155 | | | | | | |
| $\sigma^2_U$ | 1.079 | 0.205 | 4.082 | | | | | | |
| $\Psi_{W0}$ | 6.436 | -6.158 | 20.275 | -13.780 | -17.245 | -10.183 | -1.280 | -11.875 | 9.808 |
| $\Psi_{W1}$ | 0.223 | -0.066 | 0.539 | -0.289 | -0.364 | -0.211 | -0.005 | -0.232 | 0.235 |
| $\Psi_{W2}$ | -0.141 | -0.393 | 0.088 | 0.170 | 0.093 | 0.249 | 0.399 | 0.166 | 0.647 |
| $\Psi_{M0}$ | -1.501 | -5.328 | 2.405 | 1.381 | -0.411 | 3.153 | -0.478 | -3.072 | 2.044 |
| $\Psi_{M1}$ | -0.033 | -0.120 | 0.052 | 0.036 | -0.002 | 0.074 | -0.008 | -0.063 | 0.046 |
| $\Psi_{M2}$ | 0.024 | -0.057 | 0.110 | -0.020 | -0.060 | 0.019 | -0.024 | -0.076 | 0.028 |
| $\Psi_{U0}$ | -1.256 | -10.517 | 7.155 | | | | | | |
| $\Psi_{U1}$ | 0.033 | -0.165 | 0.227 | | | | | | |
| $\Psi_{U2}$ | 0.106 | -0.137 | 0.315 | | | | | | |
| $\delta$ | | | | 3.168 | 3.110 | 3.215 | | | |
| $\theta$ | | | | 3.505 | 3.415 | 3.590 | | | |
| $f$ | | | | 0.00276 | 0.00275 | 0.00277 | | | |





**Table 6.** Summary of the shape parameter estimates $\eta_j$ of the Weibull intensity function, for $j = 1, \ldots, 20$, obtained by Models A, B, and C.

| Station | Model A | | | Model B | | | Model C | | |
|---|---|---|---|---|---|---|---|---|---|
| | mean | 2.5 | 97.5 | mean | 2.5 | 97.5 | mean | 2.5 | 97.5 |
| $s_1$ | 0.838 | 0.782 | 0.891 | 0.876 | 0.874 | 0.881 | 0.989 | 0.960 | 1.028 |
| $s_2$ | 0.838 | 0.781 | 0.890 | 0.875 | 0.873 | 0.879 | 0.989 | 0.959 | 1.028 |
| $s_3$ | 0.830 | 0.774 | 0.888 | 0.887 | 0.886 | 0.893 | 0.994 | 0.966 | 1.036 |
| $s_4$ | 0.848 | 0.787 | 0.903 | 0.862 | 0.861 | 0.869 | 0.998 | 0.972 | 1.030 |
| $s_5$ | 0.828 | 0.772 | 0.885 | 0.876 | 0.873 | 0.882 | 1.012 | 0.984 | 1.046 |
| $s_6$ | 0.882 | 0.811 | 0.955 | 0.826 | 0.824 | 0.831 | 1.001 | 0.967 | 1.028 |
| $s_7$ | 0.871 | 0.804 | 0.934 | 0.839 | 0.836 | 0.848 | 0.989 | 0.958 | 1.015 |
| $s_8$ | 0.815 | 0.764 | 0.882 | 0.890 | 0.889 | 0.894 | 1.014 | 0.991 | 1.047 |
| $s_9$ | 0.812 | 0.763 | 0.876 | 0.887 | 0.886 | 0.891 | 1.012 | 0.986 | 1.046 |
| $s_{10}$ | 0.825 | 0.772 | 0.887 | 0.871 | 0.869 | 0.878 | 1.025 | 1.000 | 1.058 |
| $s_{11}$ | 0.820 | 0.764 | 0.875 | 0.865 | 0.863 | 0.869 | 1.018 | 0.987 | 1.045 |
| $s_{12}$ | 0.832 | 0.770 | 0.886 | 0.863 | 0.861 | 0.871 | 1.007 | 0.978 | 1.038 |
| $s_{13}$ | 0.868 | 0.801 | 0.929 | 0.837 | 0.833 | 0.844 | 1.010 | 0.973 | 1.043 |
| $s_{14}$ | 0.777 | 0.705 | 0.867 | 0.921 | 0.920 | 0.925 | 1.040 | 1.005 | 1.076 |
| $s_{15}$ | 0.834 | 0.778 | 0.894 | 0.876 | 0.873 | 0.883 | 1.024 | 0.992 | 1.059 |
| $s_{16}$ | 0.807 | 0.756 | 0.887 | 0.908 | 0.906 | 0.913 | 1.045 | 1.023 | 1.073 |
| $s_{17}$ | 0.803 | 0.744 | 0.889 | 0.913 | 0.911 | 0.921 | 1.053 | 1.018 | 1.089 |
| $s_{18}$ | 0.812 | 0.763 | 0.896 | 0.906 | 0.903 | 0.911 | 1.046 | 1.021 | 1.075 |
| $s_{19}$ | 0.822 | 0.769 | 0.901 | 0.894 | 0.893 | 0.898 | 1.043 | 1.009 | 1.086 |
| $s_{20}$ | 0.798 | 0.703 | 0.874 | 0.936 | 0.934 | 0.939 | 0.969 | 0.894 | 1.049 |





**Table 7.** Summary of the scale parameter estimates $\gamma_j$ of the Weibull intensity function, for $j = 1, \ldots, 20$, obtained by Models A, B, and C.

| Station | Model A | | | Model B | | | Model C | | |
|---|---|---|---|---|---|---|---|---|---|
| | mean | 2.5 | 97.5 | mean | 2.5 | 97.5 | mean | 2.5 | 97.5 |
| $s_1$ | 0.062 | 0.042 | 0.085 | 0.178 | 0.173 | 0.182 | 0.081 | 0.061 | 0.098 |
| $s_2$ | 0.061 | 0.041 | 0.084 | 0.181 | 0.175 | 0.187 | 0.085 | 0.062 | 0.108 |
| $s_3$ | 0.065 | 0.043 | 0.091 | 0.168 | 0.163 | 0.175 | 0.082 | 0.059 | 0.100 |
| $s_4$ | 0.056 | 0.040 | 0.076 | 0.201 | 0.190 | 0.209 | 0.076 | 0.061 | 0.091 |
| $s_5$ | 0.061 | 0.042 | 0.082 | 0.178 | 0.172 | 0.184 | 0.068 | 0.055 | 0.081 |
| $s_6$ | 0.042 | 0.026 | 0.065 | 0.283 | 0.270 | 0.298 | 0.078 | 0.063 | 0.103 |
| $s_7$ | 0.044 | 0.028 | 0.062 | 0.254 | 0.242 | 0.264 | 0.089 | 0.073 | 0.112 |
| $s_8$ | 0.069 | 0.044 | 0.092 | 0.156 | 0.150 | 0.161 | 0.055 | 0.043 | 0.066 |
| $s_9$ | 0.068 | 0.044 | 0.090 | 0.159 | 0.154 | 0.163 | 0.055 | 0.043 | 0.065 |
| $s_{10}$ | 0.061 | 0.042 | 0.080 | 0.184 | 0.176 | 0.193 | 0.063 | 0.050 | 0.074 |
| $s_{11}$ | 0.060 | 0.042 | 0.090 | 0.195 | 0.187 | 0.202 | 0.054 | 0.044 | 0.068 |
| $s_{12}$ | 0.058 | 0.041 | 0.087 | 0.197 | 0.186 | 0.208 | 0.046 | 0.037 | 0.056 |
| $s_{13}$ | 0.045 | 0.032 | 0.066 | 0.257 | 0.242 | 0.273 | 0.065 | 0.050 | 0.085 |
| $s_{14}$ | 0.088 | 0.050 | 0.138 | 0.115 | 0.111 | 0.119 | 0.032 | 0.023 | 0.040 |
| $s_{15}$ | 0.062 | 0.042 | 0.089 | 0.175 | 0.162 | 0.185 | 0.039 | 0.030 | 0.048 |
| $s_{16}$ | 0.079 | 0.047 | 0.108 | 0.131 | 0.123 | 0.137 | 0.032 | 0.026 | 0.038 |
| $s_{17}$ | 0.080 | 0.047 | 0.114 | 0.125 | 0.117 | 0.132 | 0.028 | 0.021 | 0.036 |
| $s_{18}$ | 0.077 | 0.046 | 0.103 | 0.134 | 0.129 | 0.140 | 0.033 | 0.026 | 0.038 |
| $s_{19}$ | 0.071 | 0.043 | 0.098 | 0.148 | 0.143 | 0.154 | 0.032 | 0.023 | 0.042 |
| $s_{20}$ | 0.098 | 0.053 | 0.173 | 0.103 | 0.098 | 0.106 | 0.075 | 0.039 | 0.130 |





**Table 8.** Summary of the estimates for the parameters $\alpha_j$ and $\beta_j$ of the function $\mu$ in the Hawkes process, for $j = 1, \ldots, 20$, obtained for Model A.

| | $\alpha$ | | | $\beta$ | | |
|---|---|---|---|---|---|---|
| Station | mean | 2.5 | 97.5 | mean | 2.5 | 97.5 |
| $s_1$ | 0.049 | 0.042 | 0.057 | 0.064 | 0.053 | 0.078 |
| $s_2$ | 0.049 | 0.042 | 0.057 | 0.062 | 0.051 | 0.074 |
| $s_3$ | 0.049 | 0.042 | 0.057 | 0.063 | 0.052 | 0.075 |
| $s_4$ | 0.049 | 0.042 | 0.057 | 0.064 | 0.053 | 0.078 |
| $s_5$ | 0.048 | 0.041 | 0.056 | 0.060 | 0.050 | 0.073 |
| $s_6$ | 0.046 | 0.038 | 0.054 | 0.057 | 0.047 | 0.069 |
| $s_7$ | 0.048 | 0.039 | 0.057 | 0.059 | 0.048 | 0.071 |
| $s_8$ | 0.044 | 0.037 | 0.053 | 0.057 | 0.047 | 0.072 |
| $s_9$ | 0.045 | 0.037 | 0.053 | 0.058 | 0.047 | 0.073 |
| $s_{10}$ | 0.048 | 0.041 | 0.058 | 0.060 | 0.050 | 0.073 |
| $s_{11}$ | 0.049 | 0.040 | 0.064 | 0.067 | 0.054 | 0.085 |
| $s_{12}$ | 0.040 | 0.033 | 0.048 | 0.056 | 0.045 | 0.071 |
| $s_{13}$ | 0.045 | 0.037 | 0.053 | 0.057 | 0.046 | 0.070 |
| $s_{14}$ | 0.040 | 0.032 | 0.050 | 0.059 | 0.045 | 0.077 |
| $s_{15}$ | 0.037 | 0.028 | 0.045 | 0.053 | 0.040 | 0.069 |
| $s_{16}$ | 0.037 | 0.030 | 0.046 | 0.057 | 0.044 | 0.074 |
| $s_{17}$ | 0.035 | 0.028 | 0.044 | 0.055 | 0.041 | 0.074 |
| $s_{18}$ | 0.037 | 0.030 | 0.045 | 0.058 | 0.045 | 0.078 |
| $s_{19}$ | 0.035 | 0.028 | 0.044 | 0.053 | 0.040 | 0.071 |
| $s_{20}$ | 0.046 | 0.034 | 0.060 | 0.067 | 0.048 | 0.092 |





This study proposed a novel geostatistical model based on self-exciting Hawkes processes for modeling the R20mm climate index, representing an innovative extension of the class of non-homogeneous spatio-temporal Poisson models. The main motivation for developing the model stems from the empirical observation that extreme rainfall events in northern Maranhão tend to occur in clusters, especially during the rainy season, suggesting temporal dependence between events.

    The proposed model incorporates temporal dependence through an excitation function and spatial dependence via hierar-
360 chical Gaussian processes, allowing for interpolation at locations with no observed data. Parameter estimation was conducted under a Bayesian framework using Markov Chain Monte Carlo (MCMC) methods. The model's predictive performance was assessed through an extensive cross-validation study, comparing the results to those obtained from Poisson models with and without seasonality.

    The results indicate that the Hawkes-based model outperformed the competing models in terms of predictive accuracy,
particularly in regions with pronounced rainfall seasonality. Additionally, the excitation function parameters provided further insights into the intensity and persistence of extreme events, revealing spatio-temporal patterns not adequately captured by conventional models.

    We conclude that the proposed model is promising for applications in climatology, especially in regions with high spatio-temporal rainfall variability. It contributes to the improvement of climate extremes analysis and forecasting, with potential
applications in the planning of climate change adaptation strategies and natural disaster mitigation.

## 6    Conclusions

This study proposed a novel geostatistical model based on self-exciting Hawkes processes for modeling the R20mm climate index, representing an innovative extension of the class of non-homogeneous spatio-temporal Poisson models. The main motivation for developing the model stems from the empirical observation that extreme rainfall events in northern Maranhão tend
to occur in clusters, especially during the rainy season, suggesting temporal dependence between events.

    The proposed model incorporates temporal dependence through an excitation function and spatial dependence via hierarchical Gaussian processes, allowing for interpolation at locations with no observed data. Parameter estimation was conducted under a Bayesian framework using Markov Chain Monte Carlo (MCMC) methods. The model's predictive performance was assessed through an extensive cross-validation study, comparing the results to those obtained from Poisson models with and
380 without seasonality.

    The results indicate that the Hawkes-based model outperformed the competing models in terms of predictive accuracy, particularly in regions with pronounced rainfall seasonality. Additionally, the excitation function parameters provided further insights into the intensity and persistence of extreme events, revealing spatio-temporal patterns not adequately captured by conventional models.

We conclude that the proposed model is promising for applications in climatology, especially in regions with high spatio-temporal rainfall variability. It contributes to the improvement of climate extremes analysis and forecasting, with potential applications in the planning of climate change adaptation strategies and natural disaster mitigation.



*Code availability.* The spatio-temporal non-homogeneous Poisson models are implemented in the `STprocpoisson` R package (https://doi.org/10.5281/zenodo.15651335 Projeto-CNPq-Clima, 2024b), while the proposed Hawkes-based models are implemented in the

390 `STprocHawkes` R package (https://doi.org/10.5281/zenodo.15652279 Projeto-CNPq-Clima, 2024a).

*Data availability.* The data utilized in this article are freely accessible. Specifically, we analysed data collected from the SISDAGRO (Agricultural Decision Support System) platform, developed by INMET (the National Institute of Meteorology, Brazil) and the National Water and Basic Sanitation Agency (ANA). To ensure full reproducibility, the dataset is provided in both the `STprocHawkes` R package (https://doi.org/10.5281/zenodo.15652279 Projeto-CNPq-Clima, 2024a) and the `STprocpoisson` R package

(https://doi.org/10.5281/zenodo.15651335 Projeto-CNPq-Clima, 2024b).

*Author contributions.* FECM conceptualized the proposed model and coordinated the team in developing and implementing it in the R software environment. AMBN and MSP developed the statistical properties of the model and assisted in constructing the interpolation method, as well as conducting the literature review on statistical models relevant to the study. DTR contributed to the preprocessing of precipitation data for the state of Maranhão, Brazil, supported the conceptual development of the model by advising on appropriate assumptions for ex-

400 treme event analysis based on her expertise in climate extremes, and helped interpret the results from a climate science perspective. CMA implemented the model code in R and organized it into an R package format.

*Competing interests.* No competing interest is declared.

*Acknowledgements.* We deeply thank the National Council for Scientific and Technological Development (CNPq) and the Ministry of Science, Technology and Innovation (MCTI) for the generous funding granted to the project, identified by process number 405750/2022-6,

included in the so-called Call 59/2022 - Line 1 - Modelling the Global Climate System, Impacts, Vulnerability and Adaptation to Climate Change and Monitoring and Forecasting Natural Disasters. The trust and investment of these agencies were fundamental to the success of this work.

We express our gratitude to the LABEST and BME laboratories of the Statistics Department of the Federal University of Rio Grande do Norte. We thank you for generously granting access to and for permission to use your computer laboratories to implement the computational

part of this project. The collaboration and support of these laboratories were essential for the successful completion of this research.



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
