# Peer review of "Improvement of the Rnnmm type climate index approach with a spatio-temporal model based on the Hawkes process"

_EGUsphere, 2025_

## Referee Comment (RC1)

Review of the paper entitled "Improvement of the Rnnmm type climate index approach with a spatio-temporal model based on the Hawkes process" by Fidel Ernesto Castro Morales et al.

General comments

The paper proposes a geostatistical model based on self-exciting Hawkes processes for modeling the Rnnmm-type extreme climate index. It provides a detailed introduction to the proposed Hawkes process model. The performance of the model is evaluated through extensive cross-validation, with comparisons to Poisson models both with and without seasonality. The results are valuable for advancing research in the analysis and forecasting of climate extremes. The overall logic of the paper is systematic and complete. I provide several suggestions for improvement in the following sections. I recommend the paper for acceptance after minor revisions.

Specific comments

1) Page 14, Line 306: MAD and MSE first appear in Sect. 5.1.1, but their definitions are only provided in Sect. 5.1.2 (page 18, lines 316–319). I suggest moving the definitions to Sect. 5.1.1 where these abbreviations first appear. If the meanings of MAD and MSE in Sect. 5.1.1 differ from those in Sect. 5.1.2, I recommend using different abbreviations to avoid confusion.

2) Page 14, Lines 308–309: In the sentence "The results indicate that the combination of $R_3$ with $P_1$ yielded the lowest MAD and MSE values in most cases," Although this conclusion can be quantitatively supported by Tables 1–3, I further suggest including statistical values across all stations in the Tables or in the text. This would allow readers to more easily compare the performance of model A with different radius using a single representative value (e.g., mean or median MAD and MSE for all stations) that reflects overall performance across all stations.

3) Page 19, Table 4: I suggest presenting the statistical data from Table 4 in the form of a map plot. The statistical values could be displayed using colored scatter points overlaid on the map, enabling readers to intuitively observe the spatial performance of the proposed model. You could also add a discussion on whether the model performs better at lower or higher elevations, or over flat versus complex terrain. The additional figure could include six subplots arranged in a 3 × 2 layout: subplot (3,2,1) showing MAD for Model A, subplot (3,2,2) showing MSE for Model A, subplot (3,2,3) showing MAD for Model B, subplot (3,2,4) showing MSE for Model B, subplot (3,2,5) showing MAD for Model C, and subplot (3,2,6) showing MSE for Model C.

4) Finally, from a practical application perspective, I suggest presenting results that demonstrate when extreme precipitation is likely to occur in Maranhão based on your modeled Rnnmm-type extreme climate index. For example, you could indicate which

months, seasons, or years tend to experience extreme precipitation. If this is not feasible, you could alternatively provide a time series plot showing both the Rnnmm-type extreme climate index and the observed number of extreme precipitation events, with the left y-axis representing the Rnnmm-type extreme climate index and the right y-axis representing the observed extreme precipitation events counts, x-axis represent the time series. I believe this would add significant value by highlighting the practical contributions of your work to improving climate extremes forecasting.

Technical corrections
5)Page 1, Manuscript Title: Consider changing "Rnnmm type" to "Rnnmm-type" for consistency with "Rnnmm-type extreme climate index" as shown in line 2 of the same page.

6) Page 2, Line 48: The text does not provide the full form of the abbreviation IMERG. The authors should also include the full form of IMERG when it first appears on this line.

7) Page 16–17, Tables 2–3: The abbreviation MSA in Tables 2 and 3 appears to be a typographical error; please correct it to MAD.

8) Page 20 Figure 2, you should clearly point it out that which axis is the Estimated function $\Lambda_3(t)$, since your plot does not show title and unit for x-axis and y-axis. I suggest you add a title and unit (if possible) for the x-axis and y-axis. Addtionaly, the fontsize for xticks and yticks is too small, you should increase the fontsize.

---

## Referee Comment (RC2)

Referee report on 'Improvement of the Rnnmm type climate index approach with a spatio-temporal model based on the Hawkes process' submitted to EGUsphere with manuscript ID egusphere-2025-2542

My following comments are a mix of major and minor comments.

1. Minor: In the title, "Rnnmm type" should be "Rnnmm-type".

2. Major: While the article addresses an environmental statistics problem, the main contribution appears to be on the statistical side. However, the introduction cites only a few papers, mostly by the authors, and attempts to convince the reader that the existing literature is inappropriate for the problem discussed in the paper. Near the end of the Section, the authors simply claim to introduce an innovative self-exciting Hawkes process, without citing any papers or providing a proper literature review of their proposal. The introduction suggests the authors are introducing the self-exciting Hawkes process for the first time. The approach was proposed by Hawkes in the Biometrika paper "Spectra of some self-exciting and mutually exciting point processes" in 1971, more than fifty years ago.

   A clearly written paragraph including an appropriate literature survey on the self-exciting Hawkes process must be provided in the introduction section. Not only the Hawkes process, but also its spatio-temporal versions are common in the literature. For example, a review article on this topic is "A Review of Self-Exciting Spatio-Temporal Point Processes and Their Applications", by Alex Reinhart, published in Statistical Science in 2018.

   The authors should clearly indicate what their novel contribution is from a statistical perspective, or they should simply demonstrate the usefulness of an existing statistical method in the context of climate extremes.

3. Minor: The full form of IMERG is not introduced.

4. Major: Overall, the proposed model is a latent Gaussian model, where separate self-exciting Hawkes processes are used to model individual time series across locations, and then the potentially transformed spatially varying coefficients are modeled using Gaussian processes. In this approach, conditioning on the model coefficients, the data are modeled spatially independently. However, a convolution through a Gaussian process does not introduce extremal dependence. The authors can refer to a large statistical literature on spatial extremes in this regard. Hence, as a spatiotemporal model for inferring spatially varying coefficients, this approach may be better suited, but may not be from a spatial-extreme perspective.

5. Major: The authors choose gamma priors for the variance-related hyperparameters, while an inverse-gamma prior would be conjugate. The justification for choosing a non-conjugate prior should be provided. In the algorithm, the authors mention Step 2 as GI. What does it mean? Inverse-gamma? If so, the usual notation is IG.

6. Major: The manuscript does not include details on the MCMC diagnostics. Besides, no simulation study has been shown. While I agree that EGUsphere is an environmental sciences-focused journal, such details should be provided in the supplement, as the main focus of the manuscript is statistical modeling. Given that there are only 20 locations, I am highly curious about the MCMC chains for the spatial dependence parameters.

7. Minor: I feel that presenting some tables in a horizontal fashion rather than in a vertical fashion (like now) would look better.

8. Minor: Although the authors claim to provide an extensive cross-validation analysis, my concern is that they draw this conclusion based on a very limited dataset. Given that no extensive simulation has been shown, I think the word "extensive" should be toned down unless a larger number of spatial locations or a larger spatial domain is used.